# Influence of Stress Assessed through Infrared Thermography and Environmental Parameters on the Performance of Fattening Rabbits

**DOI:** 10.3390/ani11061747

**Published:** 2021-06-11

**Authors:** Juan Antonio Jaén-Téllez, María José Sánchez-Guerrero, Mercedes Valera, Pedro González-Redondo

**Affiliations:** Departamento de Agronomía, Universidad de Sevilla, Ctra. Utrera km 1, 41013 Seville, Spain; jantonio.jaen@juntadeandalucia.es (J.A.J.-T.); mvalera@us.es (M.V.); pedro@us.es (P.G.-R.)

**Keywords:** animal welfare, fattening performance, infrared thermography, rabbit, stress, temperature discipline

## Abstract

**Simple Summary:**

The aim of this study was to evaluate the impact of stress due to heat (temperature-humidity index; THI) or handling (human restraining), assessed using infrared thermography, on the performance parameters of rabbits of a Spanish Common breed. Thirty-nine rabbits weaned at the age of 28 days were analyzed during a 38-d fattening period at two times of the year: a cold period and a warm period. The rabbits’ stress due to handling was assessed by the temperature difference taken by infrared thermography in the inner ear of the animals, before and after being handled. In general, the productive results were low, since it was an unimproved rustic breed. The animals were more productive in the cold season as the values obtained for daily feed intake (DFI), average daily gain (ADG), total body weight (TBW), total feed intake (TFI) and total weight gain (TWG) were higher then, while the feed conversion ratio (FCR) was higher in the warm season. The greater the stress due to handling, the less efficient the animals were. It was therefore concluded that changes in animal welfare caused by the rabbits’ reactivity to both climatic and individual factors affect animal productivity.

**Abstract:**

Rabbits often experience stress when they perceive certain stimuli, such as handling. The physiological response of animals to stress and temperament is associated with feeding efficiency, with those with the least capacity to react to stress showing the highest performance. The aim of this study was to evaluate the impact of stress due to heat (temperature-humidity index; THI) or handling, assessed using infrared thermography, on the performance parameters of rabbits of a Spanish Common breed. Thirty-nine rabbits were analyzed during a 38-d fattening period at two times of the year: a cold period and a warm period. The rabbits’ stress due to handling was assessed by the temperature difference taken in the inner ear, before and after being handled. The animals were more productive in the cold season. Temperature-humidity index significantly influenced average daily gain (ADG) and daily feed intake (DFI). Rabbits with higher levels of stress showed higher ADG, DFI and feed conversion ratio (FCR) values. The greater the stress due to handling, the less efficient the animals were. FCR increased with higher THI. It was concluded that changes in animal welfare caused by the rabbits’ reactivity to both climatic and individual factors affect animal productivity.

## 1. Introduction

Docility and not being afraid of people are among the preferential characteristics selected when domesticating animals [1]. Rabbits, however, were domesticated much later than other species [2] and the effects of domestication on them are not so marked: in fact, they conserve many characteristics of their wild ancestors, such as digging burrows and making nests [3,4]. Another of these is that domestication has failed to eradicate their fear of humans, who are perceived as a potential predator [5]. Nevertheless, proper treatment by breeders can decrease the animals’ fear and contribute to their welfare [6]. Animals often experience stress when they perceive certain stimuli, such as handling by breeders, and it is possible to differentiate between acute and chronic stress [7]. Correct handling has extremely positive effects in this species, especially if it starts soon after birth [8], since the critical period for socialization occurs when the kits first open their eyes in the nest, at around 10 days of age [9].

In addition, the environmental conditions, especially temperature and relative humidity, affect both the physiological state and the productivity of animals [10]. Rabbits are particularly sensitive to heat, as they have few sweat glands and find it hard to reduce their body heat [11,12]. The rabbit has a thermoneutral zone between 18 and 21 °C, in which it makes no effort to raise or reduce its temperature [13,14] and in which it neither suffers any stress from trying to maintain its body temperature nor requires any extra energy to regulate it to the detriment of production indexes [15]. However, heat stress in rabbits does have a negative influence on production [16], with high temperatures the main factor affecting rabbit production in regions with a warm climate [17]. In this case, a parallel pathway to that of stress management is used, whereby the hypothalamus receives information about the environment and coordinates its responses through nerves and hormones.

Physiological changes occur in order to cope with this stimulus [18]. In the short term, a stressful stimulus response is triggered by activation of the sympathetic nervous system, which releases catecholamines (norepinephrine and epinephrine) from the adrenal medulla [19]. According to Axelrod and Reisine [20], motor activity and heart and spleen contractions increase in this phase, releasing more red blood cells. Vasomotor adjustments and dilation of the pupil occur, blood clotting increases and lymphocytes are neutralized to repair tissue damage, while energy is increased through glycolysis and lipolysis [20].

Animals may have different behavioral responses to handling; this is what is called temperament [21]. Animals can be more easily stressed if they are more excitable [22]. Furthermore, the ability of each animal to cope with stressful and adverse situations is different [23,24,25]. Physiological changes caused by stress can have adverse effects on rabbits’ productive performance, affecting parameters such as feed consumption, increased body weight, feed conversion and the quality of the meat obtained [26]. The physiological response of animals to stress and temperament are associated with feeding efficiency, with those animals with the least capacity to react to stress being the most effective [27]. These adverse effects have a bearing on the viability of commercial production of rabbit meat, since the fattening period accounts for at least 40% of the animal’s lifetime on the farm [28] and a large part of farm production costs (around 40%) is spent on feeding during the fattening period [29,30].

Stress can be estimated through biomarkers and changes in body temperature, and these can be analyzed using different techniques, such as by analyzing the biochemical parameters in the blood [31] and in excreta [32], or by infrared thermography [33,34]. Changes in body surface temperature as a consequence of both acute and chronic stress can be detected by infrared thermography [35,36]. Electromagnetic energy is measured using an infrared camera, and detects the different wave frequencies emitted by each temperature value [35]. Its main strength is that it is a non-invasive and non-contact method which can measure the surface temperature of animals [37]. It has been used successfully in ruminants [35], pigs [38], dogs [39], horses [40] and rabbits [33,41].

As has been shown, handling and environmental conditions (temperature and humidity) can be two of the most common stressors in current rabbit production systems. No record has been found of any studies describing how heat stress (measured with a temperature-humidity index) and stress caused by handling (animals measured with infrared thermography) in the post-weaning phase can influence the performance parameters of rabbits. For this reason, the main aim of this work was to evaluate the effect of the levels of heat stress, assessed through the temperature-humidity index (THI) and handling stress, evaluated by using infrared thermography. Infrared thermography was measured in the inner ear of rabbits reared for meat production, on the performance parameters of the fattening period, during the cold and warm seasons.

## 2. Materials and Methods

### 2.1. Animals and Husbandry

Common Spanish agouti-coated domestic meat-oriented rabbits belonging to a strain kept at the Higher Technical School of Agricultural Engineering Teaching Farm of the University of Seville (Spain) were used. The genetic characterization [42] and productive performance [43] of this nucleus has been previously described. Overall, the rabbits were phenotypically similar to the recently recognized autochthonous breed “Antiguo Pardo Español” (Spanish Common Rabbit) [44].

During the experimental periods, the rabbits were individually housed in polyvalent wire-mesh cages measuring 90 × 40 × 30 cm^3^ (length, width and height, respectively), located in a conventional closed facility with natural ventilation (geographic coordinates: 37° 21′ 36.3″ N and 5° 56′ 23.9″ W; 11 m a.s.l.). The animals were subjected to a natural photoperiod. The trial was carried out in accordance with the Spanish legislation [45] and Directive 2010/63/EU on the protection of animals used for scientific purposes [46]. The experimental protocol was approved (Ref. 25583347/2015) by the Escuela Internacional de Doctorado (International Doctorate School) of the University of Seville (Spain).

The rabbits were fed a pelleted commercial balanced diet (15.0% crude protein and 15.5% crude fiber) ad libitum as their only feed. Water was supplied ad libitum.

### 2.2. Collecting the Temperature Data

A total of 39 weaned rabbits, with an age of 28 days old, were used during a 38-d fattening period in two different seasons: spring (warm season; 14 April to 22 May 2015; mean minimum temperature = 12.5 °C, mean maximum temperature = 24.6 °C; 12 days of data collection; *n* = 17 rabbits (12 males and 5 females)) and winter (cold season; 9 January to 23 February 2016; mean minimum temperature= 7.1 °C, mean maximum temperature = 18.1 °C;11 days of data collection; *n* = 22 rabbits (8 males and 14 females)). The animals’ stress levels were assessed with inner ear temperature measurements, following previous research by Jaén-Téllez et al. [33] which showed that the inner ear is the most suitable region to assess stress in rabbits, and that this correlates closely with eye temperature. Temperature samples were collected twice a week (Monday and Thursday) and twice each day: first, at 11:00 h when the animal had been undisturbed (U) in its own cage since the previous sampling day, and second, about ten minutes after handling the rabbit (held in the keeper’s arms) for about sixty seconds (H), always by the same person. The rabbits were restrained by the thorax and the hindquarters following the recommendations given by Chapel et al. [47]. Six pictures temperature image were taken for each rabbit twice a week. Three pictures were taken with the rabbit undisturbed in its own cage before the rabbit had been handled (U) and three when handling the rabbit (H). For each category of temperature (U or H), the mean between the three measurements was calculated. The whole procedure for the entire experimental stock took about 2 h 30 min on each sampling day. The temperature images were taken with a FLIR i7 camera, following the indications by Bartolomé et al. [48], adapted for recording inner ear temperatures in rabbits by Jaén-Telléz et al. [33]. The FLIR i7 has a precision of ± 2 °C o ± 2% and an accuracy of 0.10 °C. In order to calibrate the camera results, the environmental temperature and relative humidity were recorded with a digital thermo-hygrometer (Extech^®^ 44550, Waltham, MA, USA) every time an infrared temperature sample was taken, so that each infrared temperature had a corresponding humidity and room temperature reading.

### 2.3. Environmental and Infrared Temperatures

In order to evaluate the environmental conditions and their relationship with the infrared temperature, the following data were recorded:

Temperature, U (°C) = environmental temperature taken just at the moment when the infrared temperatures were taken in undisturbed rabbits inside the cage.

Relative humidity, U (%) = Relative humidity taken just at the moment when the infrared temperatures were taken in undisturbed rabbits

Inner ear DIF = Infrared inner ear temperature in the handled rabbit (H)—Infrared inner ear temperature in the undisturbed rabbit inside the cage (U).

### 2.4. Collecting the Productive Data and Calculating Fattening Performance

The 39 weaned rabbits were weighed on the sampling days after the temperature samples were collected. The weight gain by each rabbit and the weight of the feed was recorded twice each day on sample collection: once before being filled (weight of the feed remaining in the feeder) and second, when the feed was added, always after the pictures had been taken. With these measurements, the following productive traits were calculated:

Daily feed intake (DFI, g/d), calculated as:DFI = Feed weight differential (g) between two data collection dates/Days elapsed between those two data collection dates.

Average Daily Gain (ADG, g/d), calculated as:ADG = Live weight differential (g) between two data collection dates/Days elapsed between those two data collection dates.

The Feed Conversion Ratio (FCR) was calculated over two consecutive data collection days as:FCR = feed intake (g)/weight gain (g).

Total body weight (TBW), total weight gain (TWG) and total feed intake (TFI) during the entire fattening period were also calculated.

### 2.5. Temperature-Humidity Index

The temperature-humidity index (THI), an indicator of thermal comfort level for animals in a housing system, was calculated according to Marai et al. [49], and given as:THI = t − ((0.31 − 0.31 × RH) × (t − 14.4))(1)
where: t = temperature (°C) and RH = relative humidity percentage/100.

### 2.6. Statistical Analyses

The descriptive statistics (number, mean, minimum, maximum and standard error (SE)) for each trait: TBW, TWG, TFI, ADG, DFI, FCR and THI are shown in Table 1.

A General Linear Model of repeated measures was performed, considering each weaning rabbit (*n* = 39) as the experimental unit (Table 2). Daily feed intake, ADG and FCR were analyzed using the MIXED procedure [50] for repeated measures [51], assuming a composite symmetric (CS) covariance structure for animal effects, and including the fixed effects of sex, season, stress level and the effect of THI as a covariate. Stress levels were defined depending on the Inner ear DIF of the weaned rabbit at that sampling moment, following previous research by Jaén-Téllez et al. [33] which showed that the inner ear is the most suitable region to assess stress in rabbits. According to the temperature distribution, we defined four stress levels, graded in five-degree intervals that ensured sufficient sample size per interval: Level 1 (not stressed; *n* = 17): Inner ear DIF ≤ 0 °C; Level 2 (slightly stressed; *n* = 208): 0 °C < Inner ear DIF ≤ 5 °C; Level 3 (stressed; *n* = 114): 5 °C < Inner ear DIF ≤ 10 °C and Level 4 (very stressed; *n* = 56): Inner ear DIF > 10 °C. Each environmental effect was analyzed one-by-one using one-factor analyses (Table 2). This was followed by a Tukey-Kramer post-hoc test to study the categorical effects (Table 3).

Finally, in order to predict ADG, DFI and FCR based on stress levels and THI, a quadratic regression analysis was performed (Figure 1).

The quadratic equation with the coefficient of determination (R^2^), the root mean squared error (RMSE) and the p value of the analysis were provided for each parameter. Effects were considered as significant when *p* < 0.05. Statistical analyses were performed using the Statistica v.12.0 (Statistica software, v.12.0. Statsoft, Inc. 1984–2014, Palo Alto, CA, USA) and SAS [50] package.

## 3. Results

The fattening performance of the experimental stock is shown in Table 1. FCR showed a mean value of 3.18 ± 0.06 which ranged from 3.08 ± 0.09 for the cold season to 3.24 ± 0.07 for the warm season. In contrast, TWG, TBW, ADG and TFI showed higher values in the cold season than the warm season (increases of 36.8 g, 110 g, 2.6 g/d and 125 g, respectively) (Table 1). In the warm season, all the variables except THI had a lower coefficient of variation, and the rabbits gained less weight (TWG) and ate less feed (TFI). The changes in THI between summer and winter were high, with the minimum THI in the warm season (17.19) higher than the maximum value in the cold season (15.35).

The environmental effects that could most influence the fattening performance were studied (Table 2). Sex was not a significant effect in any trait. The stress levels produced statistically significant differences for all the productive traits collected (*p* < 0.05). The season (warm or cold) had a statistically significant effect on DFI and FCR (*p* = 0.007 and *p* < 0.001), and THI had a significant effect on ADG (*p* = 0.018) and DFI (*p* < 0.039).

The results of the Tukey post hoc are shown in Table 3. The cold season was a significantly better season for DFI and FCR. The week of the fattening period significantly affected the three productive traits studied, increasing their values almost every week. The stress levels significantly affected the three productive traits studied. The rabbits with the lowest stress levels had lower values for the three productive traits studied.

The parameters depend on the season, warm or cold, with a DFI of 76.81 g/d in summer and 80.37 g/d in winter, which leads to a higher feed consumption in the cold season, and an ADG of 24.96 g/d in summer and 27.39 g/d in winter, which increases the rabbits’ weight in the cold season. The FCR is 3.29 in summer and 3.22 in winter, which therefore makes it a better indicator in the cold season.

Depending on the level of stress, there are also changes in the productive parameters. The DFI is 44.71 g/d in non-stressed animals, but as high as 90.41 g/d in highly-stressed animals, producing a higher feed consumption the more stressed the rabbits are. The ADG ranges from 22.45 g/d in non-stressed animals to 29.14 g/d in highly-stressed animals, which results in a greater weight increase in rabbits with higher stress levels. FCR increases with the stress levels of rabbits, with 2.53 in non-stressed animals and up to 3.24 in highly-stressed animals.

The evolution of the fattening performance parameters depending on the THI and stress levels (quadratic regression analysis) is shown in Figure 1. As shown by the equations, the DFI decreased as THI increased, in contrast to stress levels. ADG was inversely proportional to THI and directly proportional to stress levels. FCR was directly proportional to THI and stress levels. Even though the three equations have been significant (*p* < 0.001), the best adjustments (R^2^ = 0.17) have been produced with the DIF and FCR variables.

## 4. Discussion

Daily feed intake, ADG and FCR are among the most essential performance parameters when evaluating productive efficiency during fattening on a rabbit farm [28]: they affect productive efficiency and are influenced by the animals’ welfare. Of these, FCR is the most important parameter from an economic point of view [29,30]. The FCR of the rabbits in this study was lower than the value obtained by González-Redondo [43] for this same breed (3.46 at 63 days of age) and slightly higher than that obtained by Feki et al. [52] in selected lines (3).

Average Daily Gain showed a lower mean value than that found by González-Redondo [43] for this strain (34.0 ± 0.93 g/day) and by Rodellar et al. [53] for Spanish Common rabbits (31.8 ± 1.77 g/day), and this value is below the normal values in select lines of young rabbits (38–48 g/d; [52]). The ADG in countries with a consolidated rabbit industry ranges from 30 to 46 g/day [54,55,56,57,58], and can be as high as 50 g/day on commercial farms which use highly prolific maternal lines and high-growth paternal lines [59,60].

Daily feed intake showed an average value (77.80 ± 1.34 g/day) lower than that of selected lines, which usually present values of around 105 g/day [58].

The low ADG and DFI values observed in the animals in this study are due to the fact that it is an unimproved rustic breed, like the Ibizan rabbit [61], and a breed with little genetic potential for feed consumption, as indicated by Ponce de León et al. [62] for the rustic Cuban semi-giant breed, and with low growth potential [63]. According to Gupta et al. [64], FCR and DFI are influenced by breed, as is ADG [65]. Lebas et al. [66] showed that the productive performance, under similar environmental and nutritional conditions, is specific to each breed, due to the genetic differences between them.

Consistent with Gupta et al. [64], we found a higher DFI in the cold season (80.30 ± 2.63 g/day) than in the warm season (76.21 ± 1.42 g/day). Similarly, we also coincided with Sabah and Dalal [67] and Gupta et al. [64], who observed seasonal differences in FCR, which was higher in the warm season than in the cold season. As in the works by Ramon et al. [68] and Ondruska et al. [69], ADG decreases as the temperature increases. Heat conditions have a negative impact on growth, due to a series of effects that can be explained through heat stress response [70]. Rabbits are highly sensitive to heat because they have very few sweat glands and these are not distributed throughout their body; they therefore have great difficulty in eliminating body heat when the ambient temperature is high and so are prone to stress [71].

The peripheral thermal receptors are triggered to induce the center of appetite in the hypothalamus to cause a decrease in DFI [72] and ADG, and an increase in FCR [71]. The high ambient temperature induces rabbits to stabilize their body heat by dissipating their latent heat, which affects the metabolic balance of water, proteins, energy and minerals, enzymatic reactions, hormonal secretions and blood metabolites [73]. During heat stress, animals increase their heat loss through the vaporization of water during respiration [71]. Heat stress not only has negative effects on growth characteristics, but also on reproduction and reduced resistance to diseases [74]. The temperature-humidity index had a significant effect on ADG and DFI, showing that THI is an optimal tool to assess the impact of heat and humidity, which often influence animals’ welfare and growth [75].

The sex of the animals had no significant effect on the productive parameters, which is normal in rabbits which have not reached sexual maturity [76], at which physiological moment the technical indices in males and females begin to diverge [77].

The handling of the animals during the fattening period had a significant influence on the productive parameters: with the increase in stress levels, there was a tendency towards increased DFI, ADG and FCR. Acute stress can induce hyperphagia with increased feed consumption [78], which turns to generate greater body weight [79]. López-Espinoza et al. [80] have also shown that there is an increase in feed consumption after the exposure to stress has ended, and in other species such as rats, feed intake acts as a comforter which reduces the stress response [81]. Nkrumah et al. [82] and Llonch et al. [83] have shown in other species that the response to handling is associated with temperament, with the most temperamental animals being the least efficient. In our trial, due to the hyperphagia produced by stress, animals which are more reactive to the effect of handling consume more feed, thus increasing ADG, but not efficiently, which leads to a higher FCR. From a livestock point of view, these animals are more temperamental and with a greater capacity to react to stress. Therefore, they are not of great interest from the point of view of feeding efficiency.

Both during and just after the action of the stressful agent, there is a suppression of feed intake induced by the action of the hormones, noradrenaline [84] and corticotropin [85], which allows the animals to carry out a response. Rabbits react with flight, for which energy is required, thus inhibiting other operations of the metabolism, including the digestive system [86].

Eating in response to stress is a common means of relief and gratification in animals [87]. As a result, stress is positively associated with body weight [88]. Acute exposure to stress produces more important metabolic changes than chronic stress [89]: the increase in FCR is due to the alteration of animal welfare that negatively affects the animals’ efficiency, making them less productive [90]. Even though one of the main objectives of the domestication process is to suppress unwanted responses to extreme fear [5], domesticated animals still shy away from humans [91].

The stress level of the rabbit examined jointly with THI exerts a greater influence on DFI, ADG and FCR than THI alone. According to Oseni and Poopola [92], the overall environment could be described as comfortable, since the average THI values in both seasons were below 23. The increase in FCR with THI and the stress level of each individual show that the animals are less efficient during fattening as the THI increases and the reactivity of each animal is higher [74,90].

## 5. Conclusions

In conclusion, it has been shown that the performance parameters were altered by the action of weather conditions and the level of individual reactivity to a stressful effect, such as handling. As a result, the productive efficiency of the rabbits was higher in the cold season than in the warm season, when the lowest levels of individual reactivity were recorded. There were an increase in FCR and a decrease in ADG with increasing THI. With the increase in the ability to react to stress, there was a tendency to increase in DFI, ADG and FCR, which implies an overall decrease in the fattening performance. In addition, humidity, temperature and individual reactivity are suitable indicators which can be used to improve the rabbits’ welfare and production.

The use of rabbits with low reactivity and controlled climatic conditions could benefit rabbit welfare and productivity. These results could serve as a useful guide for productive management in fattening rabbits and the use of rustic breeds (which are better acclimatized to the local weather conditions) to benefit animal welfare in rabbit farming.

## Figures and Tables

**Figure 1 animals-11-01747-f001:**
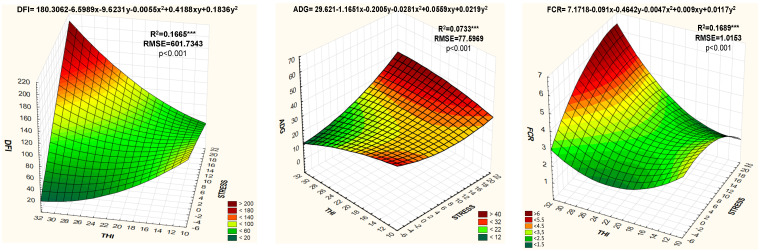
Quadratic regression analysis between stress level (*x* axis), THI (*y* axis) and DFI, ADG and FCR (*z* axis). The quadratic equation that predicts the DFI, ADG and FCR values is shown at the top of each figure, while the adjustment parameters of the R-square (R2) model, mean square error (RMSE) and *p*-value, are shown on the top right-hand side of each figure. DFI: Daily feed intake, ADG: Average daily gain, FCR: Feed conversion ratio, THI: Temperature-humidity index. *** *p* < 0.001.

**Table 1 animals-11-01747-t001:** Descriptive statistics of fattening performance of rabbits, in the warm and cold seasons.

	Variables	*N*	Mean ± SE	Minimum	Maximum	C.V. (%)
Both seasons	TBW (g)	39	1383.38 ± 40.42	577.30	1746.00	18.25
	TWG (g)	39	940.18 ± 30.91	350.50	1276.10	30.90
	TFI (g)	39	2956.57 ± 94.85	1390.70	3771.80	19.72
	ADG (g/d)	39	25.80 ± 0.46	1.70	58.70	35.24
	DFI (g/d)	39	77.80 ± 1.34	9.40	163.70	34.32
	FCR	39	3.18 ± 0.06	0.55	6.66	34.55
	THI	39	19.41 ± 0.26	11.34	30.29	26.42
Warm season	TBW (g)	22	1335.49 ± 52.66	577.30	1721.10	18.50
	TWG (g)	22	924.94 ± 35.60	352.80	1106.10	18.05
	TFI (g)	22	2901.87 ± 119.90	1390.70	3754.10	19.38
	ADG (g/d)	22	24.79 ± 0.52	2.83	58.70	32.66
	DFI (g/d)	22	76.21 ± 1.42	9.40	121.70	29.03
	FCR	22	3.24 ± 0.07	0.55	6.50	33.95
	THI	22	22.95 ± 0.20	17.19	30.29	13.60
Cold season	TBW (g)	17	1445.39 ± 61.36	809.30	1746.00	17.50
	TWG (g)	17	961.72 ± 17.79	350.50	1276.10	23.50
	TFI (g)	17	3027.35 ± 50.31	1499.00	3771.80	21.12
	ADG (g/d)	17	27.39 ± 0.83	1.70	49.57	37.61
	DFI (g/d)	17	80.30 ± 2.63	13.13	163.70	40.56
	FCR	17	3.08 ± 0.09	1.19	6.66	35.45
	THI	17	13.81 ± 0.09	11.34	15.35	7.71

TBW: Total body weight. TWG: Total weight gain. TFI: Total feed intake. DFI: Daily feed intake, ADG: Average daily gain, FCR: Feed conversion ratio, THI: Temperature-humidity index.

**Table 2 animals-11-01747-t002:** General lineal model of repeated measures of analyzed traits related to the behavior of fattening rabbits.

Factors	Degrees of Freedom	DFI		ADG		FCR	
		F-Test	*p*-Value	F-Test	*p*-Value	F-Test	*p*-Value
Sex	1	0.22	0.639	0.02	0.897	0.00	0.952
Stress level	3	16.38	<0.001	4.69	0.003	4.05	0.008
THI	1	7.67	0.039	12.05	0.018	5.81	0.061
Season	1	7.25	0.007	0.17	0.676	15.89	<0.001
Sex*Stress level	3	0.72	0.538	2.12	0.097	0.68	0.565
Sex*Season	1	0.17	0.678	0.76	0.385	3.29	0.071
Season*Stress level	2	0.33	0.721	0.49	0.615	0.00	0.952

DFI: Daily feed intake, ADG: Average daily gain, FCR: Feed conversion ratio, THI: Temperature-humidity index. A *p*-value lower than 0.05 is statistically significant.

**Table 3 animals-11-01747-t003:** Tukey-Kramer post-hoc least squared means test analysis of the significant environmental effects on Daily feed intake (DFI), Average daily gain (ADG) and Feed conversion ratio (FCR) in fattening rabbits.

**Season**						
	Warm season			Cold season		
DFI (g/d)	76.81 ^a^			80.37 ^b^		
ADG (g/d)	24.96 ^a^			27.39 ^a^		
FCR	3.29 ^b^			3.22 ^a^		
	**Week of Fattening Period**					
	1	2	3	4	5	6
DFI (g/d)	52.00 ^a^	60.93 ^a^	78.68 ^b^	90.98 ^b,c^	94.09 ^c^	103.75 ^b,c^
ADG (g/d)	25.88 ^a,b^	23.89 ^a^	27.75 ^a,b^	26.96 ^a,b^	23.65 ^a^	30.77 ^b^
FCR	2.34 ^a^	2.88 ^a,b^	3.12 ^b,c^	3.54 ^c,d^	4.18 ^d^	3.48 ^b,c,d^
	**Stress Level**					
	1	2		3		4
DFI (g/d)	44.71 ^a^	75.21 ^b^		81.06 ^b^		90.41 ^c^
ADG (g/d)	22.45 ^a^	26.22 ^a,b^		24.25 ^a^		29.14 ^b^
FCR	2.53 ^a^	3.16 ^a^		3.59 ^b^		3.24 ^a,b^

Level 1 (not stressed, 19 records): Inner ear DIF ≤ 0 °C; Level 2 (slightly stressed, 243 records): 0 °C < Inner ear DIF ≤ 5 °C; Level 3 (stressed, 134 records): 5 °C < Inner ear DIF ≤ 10 °C and Level 4 (very stressed, 60 records): Inner ear DIF > 10 °C DFI: Daily feed intake, ADG: Average daily gain, FCR: Feed conversion ratio, THI: Temperature-humidity index. Different superscript letters (^a–d^) indicate significant differences between the values.

## Data Availability

The data presented in this study are available on request from the corresponding author.

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
