# Peer review of "Influence of Stress Assessed through Infrared Thermography and Environmental Parameters on the Performance of Fattening Rabbits"

_animals, 2021, doi:10.3390/ani11061747_

Round 1

Reviewer 1 Report

Animals Review Jaen – Tellez 1214862

General Comment: This is an interesting study involving the non invasive measurement of environmental and handling stress in rabbits. The work is well conducted and the manuscript is well written and referenced. The findings will be of interest to biologists and to the field of rabbit production.

Specific Comments:

Abstract:

L25 – the authors use the term “calmest” animals and “ those with the least capacity to react to stress”. They also use the language ‘calmest are most effective”. This language is somewhat undefined. This could mean anything from tissue receptor density for catecholamines to an animal psychological profile. It would be helpful if the authors could more precisely and more carefully define their language here.

Introduction:

L73 – the authors suggest the primary impact of epinephrin was to inhibit insulin as a means of affecting changes in glucose levels. Epinephrin will also and significantly affect glycolysis and gluconeogenesis which will be primary means of affecting glucose levels. This should be included.

L106 – “work is” or “work was”?

Methods:

-The authors differentiate the summer and winter temperature and THI values. However, would these values still not both be within the animal’s thermal neutral zone?

-The authors used the rabbits inner ear temperature as a proxy for the eye. Why not just use the eye Tmax, Tmean and Tmin? Please explain the advantage and reason for using the inner ear?

-For the FLIR i7 camera, please provide the details for the emissivity used, the camera calibration procedures, the accuracy and precision for the camera and why this model was used compared to the other FLIR models.

Results:

-The feed conversion information is interesting and the impact the THI had on FCR. Why are the authors using FCR instead of the more commonly and currently used values for Residual Intake and Gain (RIG)?  

Discussion:

L276 – perhaps change “coinciding with” to “consistent with” or “in agreement with”?

L294 – perhaps change “show” to “demonstrated”?

L315 – the authors write that the most temperamental animals are least efficient. This contrasts  with some of the primate research that has been done (example Sapolsky, Why Zebras don’t get ulcers. Holt and Comp. 1994) suggesting that a vigorous stress response if often a sound coping strategy seen for animals less recently domesticated. Could the authors suggest and discuss some reasons for this?  

References:

The manuscript is well referenced. One reference perhaps also relevant would be Grignaschi and Arello. 2013. In the Text “Thermography” By Luzi et al. Brescia.

Author Response

Animals Review Jaen – Tellez 1214862

General Comment: This is an interesting study involving the non invasive measurement of environmental and handling stress in rabbits. The work is well conducted and the manuscript is well written and referenced. The findings will be of interest to biologists and to the field of rabbit production.

Authors’ response: Dear Reviewer 1:

 We have taken into account all your recommendations. We would like to thank you for your useful comments, which have enabled us to significantly improve our original manuscript. Our explanations about how we have addressed these corrections can be seen in the following lines and in a corrected version of the manuscript with the changes/corrections highlighted in red.

Specific Comments:

Abstract:

L25 – the authors use the term “calmest” animals and “those with the least capacity to react to stress”. They also use the language ‘calmest are most effective”. This language is somewhat undefined. This could mean anything from tissue receptor density for catecholamines to an animal psychological profile. It would be helpful if the authors could more precisely and more carefully define their language here.

Authors’ response: These vague terms have been removed and the sentences have been modified ( L24).

Introduction:

L73 – the authors suggest the primary impact of epinephrin was to inhibit insulin as a means of affecting changes in glucose levels. Epinephrin will also and significantly affect glycolysis and gluconeogenesis which will be primary means of affecting glucose levels. This should be included.

Authors’ response: Following the recommendation of the other reviewer, these sentences has been deleted.

L106 – “work is” or “work was”?

Authors’ response: This has been changed (L99).

Methods:

-The authors differentiate the summer and winter temperature and THI values. However, would these values still not both be within the animal’s thermal neutral zone?

Authors’ response: In small animals, the THI values obtained are classified as follows: <27.8=absence of heat stress, 27.8 to <28.9= moderate heat stress, 28.9 to <30.0= severe heat stress and 30.0 and more = very severe heat stress. (Alnaimy Habeeb et al., 2018). Besides, Marai et al (2002) consider that in a THI < 27.85 no heat stress occurs in rabbits, which would justify the absence of a significant influence of THI on DFI, added to the fact that in our study we used a rustic Spanish breed of rabbit, which may therefore have a greater tolerance to heat.So, the mean THI value in the warm season was within the animal’s thermal neutral zone but, at some times, it could suffer heat stress. This did not happen in the cold season.

-The authors used the rabbits inner ear temperature as a proxy for the eye. Why not just use the eye Tmax, Tmean and Tmin? Please explain the advantage and reason for using the inner ear?

Authors’ response: This measurement has the highest correlation with acute stress, following the previous paper by Jaén-Téllez et al., 2020

-For the FLIR i7 camera, please provide the details for the emissivity used, the camera calibration procedures, the accuracy and precision for the camera and why this model was used compared to the other FLIR models.

Authors’ response: The camera has a precision of ±2 °C o ±2% and an accuracy of 0.10°C. We calibrated it with the humidity and temperature for each record and the emissivity used was 0.80 (recommended by Allan Lee Schaefer) (L144-145)

Results:

-The feed conversion information is interesting and the impact the THI had on FCR. Why are the authors using FCR instead of the more commonly and currently used values for Residual Intake and Gain (RIG)?  

Authors’ response: We are using FCR instead of other indexes (such as RIG) because it is the usual one for assessing feed efficiency in rabbits. RIG is rarely used in rabbits (as can be seen by a literature search). Moreover, RIG is not mentioned in the harmonization document on nutrition experiments in rabbits: Fernández-Carmona, J., Blas, E., Pascual, J. J., Maertens, L., Gidenne, T., Xiccato, G., & García, J. (2005). Recommendations and guidelines for applied nutrition experiments in rabbits. World Rabbit Science, 13(4), 209-228. 

Discussion:

L276 – perhaps change “coinciding with” to “consistent with” or “in agreement with”?

Authors’ response: This has been changed (L289).

L294 – perhaps change “show” to “demonstrated”?

Authors’ response: Following the recommendation of the other reviewer, these sentences have been deleted.

L315 – the authors write that the most temperamental animals are least efficient. This contrasts  with some of the primate research that has been done (example Sapolsky, Why Zebras don’t get ulcers. Holt and Comp. 1994) suggesting that a vigorous stress response if often a sound coping strategy seen for animals less recently domesticated. Could the authors suggest and discuss some reasons for this?  

Authors’ response: In addition to the differences between species in the way of responding to and recovering from stressful situations, there is an individual differentiating factor as to why the response to potentially threatening environmental conditions of some individuals is greater when compared to the response of other individuals. These individual differences are determined by the interaction of genetic factors or with the history of behavioral interaction of the individual with its environment.The response and recovery capacity in each species must be taken into consideration, which will determine its subsequent negative influence on different parameters, including the productive ones. Specifically, in rabbits the response to stress is slow and post-stress recovery is late. In species such as rabbits, there is secondary hypertension related to high sympathetic activity (acute stress, which is what has been discussed in this article). Both considerations may be different in other species, such as primates, which differ in their responsiveness and vigor as a result of stress. Also in primates, the social rank of an animal influences stress-related physiology.In fact, Daniewski y Tadeusz Jezierski (2003) said that: "We assume that by selecting for high and low activity in the Open Field, we selected for active and passive coping style. That is, they associate high-activity rabbits in the Open Fieldtest with rabbits with an active way of coping with stimuli (which would be more temperamental), and the more temperamental ones would therefore be worse (worse fattening efficiency)

References:

The manuscript is well referenced.

One reference perhaps also relevant would be Grignaschi and Arello. 2013.

In the Text “Thermography” By Luzi et al. Brescia.

Authors’ response: We have not found the first reference and any good place for the second one.

Reviewer 2 Report

This manuscript describes the effect of stress induced by heat and handling on specific production parameters or rabbits, using infrared thermography, a technique that is currently  gaining momentum. While the topic itself is interesting and important as far as the welfare of captive and farm animals are concerned, the authors did not do themselves any favour in the way they presented their study. For instance, some elements described in the introduction are irrelevant to the study, whilst important piece of information are missing elsewhere. Also, the results should only be presented in their specific section (not in the methodology).

More specifically:

Introduction

Line 73 to 79: I doubt that much detail is needed since the study is not about hormones. To make the connection between stress, hormones and its negative effect on production parameters is one thing, but to describe the all process is another (especially within the context of this study).

Line 91 to 93: regroup with following paragraph

Line 101 to 103: references are needed. Besides, I do not think this specific topic has been discussed at any point in the introduction, yet this is quite important to give some adequate background to the rationale of the study.

Line 106: replace "is" by "was"

Line 105 to 110: the sentence is too long. Please rephrase.

Line 131: replace "analyzed" by "used"

Methods:

Line 120: The authors mentioned that the rabbits were maintained in a close facility with natural ventilation. So I am guessing that there was no additional temperature / humidity control in place? Not even during winter nights?

Line 132: I do not think April to May in this part of the world qualifies as summer. Isn't it Spring season rather? Also were all the rabbits of the same sex? if not, how many males and females were they?

Statistical analyses

I would suggest to include the id of the rabbit as a random factor in the model to control for repeated measures on the same individual.

Also, it might be worth to look at potential interactions between the different factors.

Finally, the results should not appear in the statistical section. They have to be part of the results section, to give support on the description in the text.

Results

What about the description of the effect of stress on the production parameters? This needs to be described more adequately (as in in which direction did it go? Negative effect? to what extent? etc.)

Line 247: was it significant?

Line 249 to 252: this also needs to be described in more details, and whether it was significant or not.

Discussion:

I would very much like to see the results once the elements I have described for the analyses have been included before I can assess whether the conclusions reached are actually valid.

That said, the paper cited for the argument made at line 324 to 325 refers to a study made on humans, which I do not think are relevant to farm animals. My experience of stressed animals is usually quite the opposite: they stop eating.

Author Response

This manuscript describes the effect of stress induced by heat and handling on specific production parameters or rabbits, using infrared thermography, a technique that is currently  gaining momentum. While the topic itself is interesting and important as far as the welfare of captive and farm animals are concerned, the authors did not do themselves any favour in the way they presented their study. For instance, some elements described in the introduction are irrelevant to the study, whilst important piece of information are missing elsewhere. Also, the results should only be presented in their specific section (not in the methodology).

Authors’ response: Dear Reviewer 1:

We have taken into account all your recommendations. We would like to thank you for your useful comments, which have enabled us to significantly improve our original manuscript. Our explanations about how we have addressed these corrections can be seen in the following lines and in a corrected version of the manuscript with the changes/corrections highlighted in red.

More specifically:

Introduction

Line 73 to 79: I doubt that much detail is needed since the study is not about hormones. To make the connection between stress, hormones and its negative effect on production parameters is one thing, but to describe the all process is another (especially within the context of this study).

Authors’ response: This has been changed and much detail has been deleted.

Line 91 to 93: regroup with following paragraph

Authors’ response: This has been regrouped (85-87).

Line 101 to 103: references are needed. Besides, I do not think this specific topic has been discussed at any point in the introduction, yet this is quite important to give some adequate background to the rationale of the study.

Authors’ response: References and background have been added in the introduction (60-61 and 75-76).

Line 106: replace "is" by "was"

Authors’ response: This has been changed (L99).

Line 105 to 110: the sentence is too long. Please rephrase.

Authors’ response: This has been changed (99-103).

Line 131: replace "analyzed" by "used"

Authors’ response: This has been changed (L123).

Methods:

Line 120: The authors mentioned that the rabbits were maintained in a close facility with natural ventilation. So I am guessing that there was no additional temperature / humidity control in place? Not even during winter nights?

Authors’ response: Yes, it was a thermohydrometer control; the temperature and the humidity were used in the analyses. (145-149)

Line 132: I do not think April to May in this part of the world qualifies as summer. Isn't it Spring season rather? Also were all the rabbits of the same sex? if not, how many males and females were they?

Authors’ response: This has been changed (L124).

Statistical analyses

I would suggest to include the id of the rabbit as a random factor in the model to control for repeated measures on the same individual.

Also, it might be worth to look at potential interactions between the different factors.

Authors’ response: This has been added (L185-189, L197-198, table 2 and table 3).

Finally, the results should not appear in the statistical section. They have to be part of the results section, to give support on the description in the text.

Authors’ response: This has been changed (L124).

Results

What about the description of the effect of stress on the production parameters? This needs to be described more adequately (as in in which direction did it go? Negative effect? to what extent? etc.)

Authors’ response: This has been added (240-251).

Line 247: was it significant?

Authors’ response: This has been added (231).

Line 249 to 252: this also needs to be described in more details, and whether it was significant or not.

Authors’ response: This has been added (252-258).

Discussion:

I would very much like to see the results once the elements I have described for the analyses have been included before I can assess whether the conclusions reached are actually valid.

Authors’ response: The new results have been added.

That said, the paper cited for the argument made at line 324 to 325 refers to a study made on humans, which I do not think are relevant to farm animals. My experience of stressed animals is usually quite the opposite: they stop eating.

Authors’ response: Both during the action of the stressor and immediately after, there is a suppression of feed intake induced by the action of the hormones norepinephrine and corticotropin. The action of these hormones is very short in time and they have a half-life of only a few minutes in the blood. The reaction is acute, but it is not chronic stress, which is what leads to the similarity between humans and rats.

In rats, severe stimulation can produce hypophagia by reducing the amount of food consumed in the subject, which results in low weight; softer stimulation can induce hyperphagia, an increase in food consumption, which generates obesity, (Silveira, et al., 2000; González-torres , López-Espinoza &Valeiro-Dos Santos, 2010; Dess, 1992; Dess, 1997; González-Torres, 2010).

Silveira, P., Xavier, M., Souza, F., Manoli, L., Rosat, R., Ferreira, M. & Dalmaz, C. (2000). Interaction between repeated restraint stress and concomitant midazolam administration on sweet food ingestion in rats. Brazilian Journal of Medical and Biological research, 33, 1343- 1350.

González-Torres. M., López-Espinoza. A, & Valeiro-Dos santos, C. (2010). Efecto del tipo de controlabilidad del estrés sobre la conducta alimentaria en ratas. Revista mexicana de análisis de la conducta. 36 111- 127 n 2.

Dess, N. K. (1992). Divergent responses to saccharin vs. sucroseavailability after stress in rats. Physiology&behavior, 52(1), 115-125. doi:10.1016/0031-9384(92)90440-D

Dess, N. K. (1997). Ingestion after stress: Evidencefor a shift regulatory in food- rewardedoperant performance. Learning and Motivation, 28, 342-356. doi:10.1006/lmot.1997.0974

Reviewer 3 Report

In this article, the authors show that both weather conditions and stress by handling alter the animal productivity in weaned rabbit. Weight gain during the fattening period showed higher values in the cold season than the warm season. In addition, feed conversion ratio (FCR) was directly proportional to temperature-humidity index (THI) and stress levels.

The topic addressed is interesting and deserves a constructive discussion, but I think there are some revisions that should be made before publication.

  • Line 42-43. I cannot understand what meaning of this sentence. Please revise this.
  • Line 103-105. I think there are the articles about the effect of heat stress and handling stress on rabbit. Please explain in concrete terms about the novelty.
  • Line 230-234. Please show the statistical difference.
  • Line 313-318. Were the rabbits indicating lower FCR and lower ADG above average in weight after general fattening period (not “38 days in this trial”)? Even if FCR was low, if the rabbit failed to gain weight sufficiently, then it is not possible to judge the relation of feeding efficiency and rabbit’s temperament.
  • Line 321-323. I cannot understand what meaning of this sentence. Please revise this.
  • Line 336. I cannot understand what meaning of the sentence, “due to the fact that their productivity is affected”. Please show a more detailed description.
  • Line 343. I cannot understand what meaning of the sentence, “increase in temperament”. Please revise this.

  • Line 39. Please delete the word “)”.
  • Line 62-63. Please delete the words “and controls”.
  • Line 101. Please delete the sentence “the individual reactivity of the rabbits to”.
  • Line 131, 166, 197. Please correct the word “weaning” into “weaned”.
  • Line 137. Please insert “temperature” after “ear”.
  • Line 142. Please correct the sentence “later, when the rabbit had been handled” into “after handling the rabbit”.
  • Line 145. Please insert “(temperature image)” after “pictures”.
  • Line 151, 294. Please show the number of References.
  • Line 165, 183. Please conform to the style of chapter head.
  • Table 1, “N”. I think, in ADG, DFI and FCR, “Mean” would show the mean of daily average value. Therefore, I recommend that “N” shows the number of rabbits. And I cannot understand what “N” in THI show.
  • Line 193. Please show the sentence after the word “ADG:”.
  • Line 235. Please change the sentence “all the variables had a lower coefficient of variation except THI” into “all the variables except THI had a lower coefficient of variation”.
  • Line 308. Please change the sentence “it turn generates” into “turns to generate”.
  • Line 316-318. Please change the sentence “these animals more temperamental and with a greater capacity to react to stress, are therefore not of great interest from the point of view of feeding efficiency” into “these animals are more temperamental and with a greater capacity to react to stress. Therefore, they are not of great interest from the point of view of feeding efficiency”.
  • Line 320. Please insert “, (comma)” after “hormones”.
  • Line 325. Please change the sentence “, and, as a result,” into “. As a result,”.
  • Line 342. Please correct the word “was” into “were”.
  • Line 344. Please correct the word “implying” into “which implies”.
  • Line 345-346. Please delete the word “controlling” and insert “using” after “indicators”.
  • Line 348. Please correct the word “an” into “a”.
  • Line 359. Please delete overlapped words “: (colon)”.
  • Line 362-363. Please correct the word “Internacional” into “International”.
  • Line 359. Please delete overlapped words “. (period)”.

Author Response

Authors’ response: Dear Reviewer 3:

We have taken into account all your recommendations. We would like to thank you for your useful comments, which have enabled us to significantly improve our original manuscript. Our explanations about how we have addressed these corrections can be seen in the following lines and in a corrected version of the manuscript with the changes/corrections highlighted in red.

Line 42-43. I cannot understand what meaning of this sentence. Please revise this.

Authors’ response: This has been changed (41-42).

Line 103-105. I think there are the articles about the effect of heat stress and handling stress on rabbit. Please explain in concrete terms about the novelty.

Authors’ response: The novelty has been added (96-98)

Line 230-234. Please show the statistical difference.

Authors’ response: The statistical differences were shown in Table 2 and 3: this is only a descriptive table.

Line 313-318. Were the rabbits indicating lower FCR and lower ADG above average in weight after general fattening period (not “38 days in this trial”)? Even if FCR was low, if the rabbit failed to gain weight sufficiently, then it is not possible to judge the relation of feeding efficiency and rabbit’s temperament.

Authors’ response: Ramon et al. (1996) observed in rabbits that high ADG and FCR occur in situations of heat stress, revealing poor feeding efficiency.

Line 321-323. I cannot understand what meaning of this sentence. Please revise this.

Authors’ response: This has been revised (329-330).

Line 336. I cannot understand what meaning of the sentence, “due to the fact that their productivity is affected”. Please show a more detailed description.

Authors’ response: This sentence was redundant and has been removed.

Line 343. I cannot understand what meaning of the sentence, “increase in temperament”. Please revise this.

Authors’ response: This sentence was redundant and has been removed.

Line 39. Please delete the word “)”.

Authors’ response: This has been deleted.

Line 62-63. Please delete the words “and controls”.

Authors’ response: These words have been deleted.

Line 101. Please delete the sentence “the individual reactivity of the rabbits to”.

Authors’ response: These words have been deleted.

Line 131, 166, 197. Please correct the word “weaning” into “weaned”.

Authors’ response: This has been corrected (123, 160 and 190).

Line 137. Please insert “temperature” after “ear”.

Authors’ response: This has been inserted (129).

Line 142. Please correct the sentence “later, when the rabbit had been handled” into “after handling the rabbit”.

Authors’ response: This has been corrected (134-135).

Line 145. Please insert “(temperature image)” after “pictures”.

Authors’ response: This has been inserted (137).

Line 151, 294. Please show the number of References.

Authors’ response: This has been inserted (144).

Line 165, 183. Please conform to the style of chapter head.

Authors’ response: Done.

Table 1, “N”. I think, in ADG, DFI and FCR, “Mean” would show the mean of daily average value. Therefore, I recommend that “N” shows the number of rabbits. And I cannotunderstandwhat “N” in THI show.

Authors’ response: ADG, DFI and FCR is the mean of:

DFI = weight feed differential (g) between two data collection dates / Days elapsed between those two data collection dates.

Average Daily Gain (ADG, g/d) was calculated as:

ADG = Live weight differential (g) between two data collection dates / Days elapsed between those two data collection dates.

Feed Conversion Ratio (FCR) was calculated over two consecutive data collection

days as:FCR = feed intake (g) /weight gain (g).

The n of THI is the number of THI values used in the analysis (one for each record)

Line 193. Please show the sentence after the word “ADG:”.

Authors’ response: This has been added (Table 2).

Line 235. Please change the sentence “all the variables had a lower coefficient of variation except THI” into “all the variables except THI had a lower coefficient of variation”.

Authors’ response: Thishas been changed (211-212).

Line 308. Please change the sentence “it turn generates” into “turns to generate”.

Authors’ response: This has been changed (315).

Line 316-318. Please change the sentence “these animals more temperamental and with a greater capacity to react to stress, are therefore not of great interest from the point of view of feeding efficiency” into “these animals are more temperamental and with a greater capacity to react to stress. Therefore, they are not of great interest from the point of view of feeding efficiency”.

Authors’ response: This has been changed (323-325).

Line 320. Please insert “, (comma)” after “hormones”.

Authors’ response: This has been inserted (327).

Line 325. Please change the sentence “, and, as a result,” into “. As a result,”.

Authors’ response: This has been changed (332).

Line 342. Please correct the word “was” into “were”.

Authors’ response: This has been corrected (349).

Line 344. Please correct the word “implying” into “which implies”.

Authors’ response: This has been corrected (351).

Line 345-346. Please delete the word “controlling” and insert “using” after “indicators”.

Authors’ response: This has been deleted and inserted (352-353).

Line 348. Please correct the word “an” into “a”.

Authors’ response: This has been corrected (355).

Line 359. Please delete overlapped words “: (colon)”.

Authors’ response: This has been deleted (.

Line 362-363. Please correct the word “Internacional” into “International”.

Authors’ response: It is a Spanish proper name, so we feel that “Internacional” is correct.

Line 359. Please delete overlapped words “. (period)”.

Authors’ response: This has been deleted.

Round 2

Reviewer 2 Report

I am satisfied with the changes made to the manuscript. However, standard errors associated with means need to be added throughout the manuscript and units need to be added to Table 3.

Author Response

Authors’ response: Dear Reviewers 2 and 3:

We have taken into account all your recommendations. We would like to thank again for your useful comments, which have enabled us to significantly improve our original manuscript. Our explanations about how we have addressed these corrections can be seen in the following lines and in a corrected version of the manuscript with the changes/corrections highlighted in red.

Review 2

I am satisfied with the changes made to the manuscript. However, standard errors associated with means need to be added throughout the manuscript and units need to be added to Table 3.

Contestación:

Authors’ response: This has been included (L303, L314 y Table3)

Reviewer 3 Report

Thank you for your comments and revised manuscript. I have confirmed that the manuscript has been corrected according my suggestions. However, about the following “Authors’ response”, I’m worry that the authors didn’t get my point. If rabbits are three (rabbit A, B and C), ADG of rabbit A is data “a”, ADG of rabbit B is data “b”, ADG of rabbit C is data “c”. “Mean” of ADG is the average of “a”, “b” and “c”, and “N” is three.

“Table 1, “N”. I think, in ADG, DFI and FCR, “Mean” would show the mean of daily average value. Therefore, I recommend that “N” shows the number of rabbits. And I cannot understand what “N” in THI show.”

“Authors’ response: ADG, DFI and FCR is the mean of:...”

Author Response

Authors’ response: Dear Reviewers 2 and 3:

We have taken into account all your recommendations. We would like to thank again for your useful comments, which have enabled us to significantly improve our original manuscript. Our explanations about how we have addressed these corrections can be seen in the following lines and in a corrected version of the manuscript with the changes/corrections highlighted in red.

 Review 3

Thank you for your comments and revised manuscript. I have confirmed that the manuscript has been corrected according my suggestions. However, about the following “Authors’ response”, I’m worry that the authors didn’t get my point. If rabbits are three (rabbit A, B and C), ADG of rabbit A is data “a”, ADG of rabbit B is data “b”, ADG of rabbit C is data “c”. “Mean” of ADG is the average of “a”, “b” and “c”, and “N” is three. 

“Table 1, “N”. I think, in ADG, DFI and FCR, “Mean” would show the mean of daily average value. Therefore, I recommend that “N” shows the number of rabbits. And I cannot understand what “N” in THI show.”

“Authors’ response: ADG, DFI and FCR is the mean of:...”

Contestación:

Authors’ response: Changes have been included (Table 1 )